# A Multi-Frequency Omnidirectional Antenna Based on a Ring-Shaped Structure

**DOI:** 10.3390/mi14050994

**Published:** 2023-05-03

**Authors:** Honglei Guo, Yu Chen, Qiannan Wu, Jianyang Wang, Yu He, Yonghong Cao, Mengwei Li

**Affiliations:** 1School of Instrument and Electronics, North University of China, Taiyuan 030051, China; 2The Academy for Advanced Interdisciplinary Research, North University of China, Taiyuan 030051, China; 3The Center for Microsystem Integration, North University of China, Taiyuan 030051, China; 4College of Instrumentation and Intelligent Future Technology, North University of China, Taiyuan 030051, China; 5School of Semiconductor and Physics, North University of China, Taiyuan 030051, China; 6The MOE Key Laboratory of Weak-Light Nonlinear Photonics, TEDA Applied Physics Institute and School of Physics, Nankai University, Tianjin 300457, China; 7School of Aeronautics and Astronautics, North University of China, Taiyuan 030051, China

**Keywords:** multi-frequency, microstrip antenna, split-ring resonator, defective grounding, omnidirectional radiation

## Abstract

A multi-frequency microstrip antenna loaded with a ring-like structure has been proposed. The radiating patch on the antenna surface consists of three split-ring resonator structures, and the ground plate consists of a bottom metal strip and three ring-shaped metals with regular cuts to form a defective ground structure. The proposed antenna works in six different frequency bands covering 1.10, 1.33, 1.63, 1.97, 2.08, and 2.69 GHz and works entirely when connected to 5G NR (FR1, 0.45–3 GHz), 4GLTE (1.6265–1.6605 GHz), Personal Communication System (1.85–1.99 GHz), Universal Mobile Telecommunications System (1.92–2.176 GHz), WiMAX (2.5–2.69 GHz), and other communications frequency bands. Moreover, such antennas also have stable omnidirectional radiation properties across different operating frequency bands. This antenna meets the needs of portable multi-frequency mobile devices and provides a theoretical approach for the development of multi-frequency antennas.

## 1. Introduction

In recent years, with the rapid development of wireless communication technology, wireless communication equipment is widely used in satellite navigation [1,2,3], wireless interconnection [4,5,6,7], high-speed information transmission, and other communication fields [8,9,10,11,12]. The antenna is an important component used to send and receive information in a wireless communication system [13,14]. The use of single-frequency antennas has been unable to meet the needs of modern multi-frequency communication. Moreover, when multiple antennas work at the same time in a limited space, there will be coupling effects between each other, resulting in the failure of the communication system to work normally. Modern wireless communication systems require antennas to be multifunctional, easy to integrate, low-cost, and miniaturized [15,16,17,18]. Multi-frequency antennas can work in multiple communication frequency bands at the same time. Moreover, using a single antenna to achieve multi-frequency effects, in a sense, greatly reduces the area of the antenna, saves the design cost, and meets the needs of industrial communication equipment [19,20,21]. Therefore, how to realize the multi-frequency radiation characteristics of antennae under other good conditions has become a hot research topic at present.

In recent years, various approaches have been developed to design and produce multi-frequency antennas. The main methods include the multilayer method [22,23,24], slot method [25,26], branch method [27,28], the reconfigurable method [29,30,31], the metamaterial loading method [32,33,34,35,36], and Dipole multifrequency antenna [37,38,39,40]. For example, in 2010, Chen et al. [22] proposed a dual-frequency microstrip antenna using low-temperature co-fired ceramic (LTCC) technology by adopting a multi-layer method, which can be used for GPS positioning. In 2015, Wu et al. [25] proposed a multi-frequency antenna with three operating frequency points by etching three slots of different lengths on a triangular radiation patch. In 2018, Chun [28] designed an antenna composed of multiple branches. Changing the length of the corresponding branch effectively modulates the resonance frequency and impedance matching of the antenna. This antenna is an excellent choice for LTE, WLAN, WiMAX, and other narrowband systems. In 2018, Vijayan et al. [31] proposed a new antenna that uses two PIN diodes to modify the working state of the diode without changing the physical size of the antenna, which can obtain the reconfiguration of the antenna in four different frequency bands. In 2017, Rani [33] designed a metamaterial structural unit as the radiation patch of the antenna to obtain dual-frequency performance. In 2019, Zhang et al. [34] achieved dual-frequency operation by loading a transmission line-inspired structure onto the antenna. In 2020, a four-port four-frequency antenna available at high-low frequencies was presented by Chen et al. [37]. In 2021, Wan, S.Y. et al. [38] designed a multi-frequency microstrip antenna that works at seven frequency bands. In 2022, Anupa Chatterjee et al. [39] demonstrated a four-band compact dual-element MIMO (multi-input multi-output) antenna system consisting of a compact inverted F antenna with a zigzag arm. One side of the T-type monopole antenna is extended and twisted, and the other side is grounded, forming the IFA structure. In 2023, Fatma Taher et al. [40] designed a dual-port MIMO antenna that provided broadband of 25.5–30 GHz and a peak gain of 8.75 dBi. However, the above-proposed antennas could not meet the requirements of industrial communication equipment for multi-frequency communication in terms of multi-band, size, and gain for most of the operating frequency bands of multi-frequency antennas mentioned above.

In this paper, we introduce a new multi-frequency antenna structure, which can realize the function of the multi-frequency antenna at six frequency points. This method can improve the design of multi-frequency communication equipment.

## 2. Theory and Design

The geometry of the antenna simulation model is shown in Figure 1. The antenna is printed on a 1.6 mm thick Taconic RF-60A substrate with a relative dielectric constant of 6.15 and a loss angle of 0.0028. The printed metal is copper with a thickness of 0.036 mm. Three open-loop resonator structures are loaded on the front in the form of radiation patches, as shown in Figure 1b, and circular defect ground structures with cross-shaped cross notches and rectangular ground structures are distributed in corresponding positions on the back, as shown in Figure 1c. The entire open ring resonator radiation patch structure is connected to the feeder end via a microstrip line with an impedance set to 50 Ω. The parameters of the proposed antenna are given in Table 1.

The radiation patch on the front of the antenna structure consists of three split-ring resonator structures. When an electromagnetic wave is incident on an open resonant ring, the initially stable magnetic field induced by the incident wave changes and the magnetic field induces an induced current in the metal ring. The induced current flows over the metal ring such that the metal ring has an equivalent inductance. Moreover, due to the accumulation of charges between the metal rings, the metal rings generate equivalent inductance, and *LC* resonances occur in the metal rings [41]. According to the equivalence principle of the open resonant ring, its resonant frequency can be expressed in the following formula [42].
(1)$$f_{0}=[\sqrt{(C_{SRRs}R_{SRRs}^{2})/L_{SRRs}-1}](2\pi C_{SRRs}R_{SRRs})$$

Here, the *L_SRR_* is the overall inductance of the SRRs, which is formed by the metal ring inductors in series with the current direction. *C_SRR_* is the overall capacitance of the SRRs equivalent circuit. *R_SRR_^2^* is the equivalent conductor loss, the shunt resistance. Its size is related to ring width *g*, metal thickness *t*, and metal material *ρ*.

The capacitance of the open resonant ring SRR_S_ is:(2)$$C_{SRRs}=(n-1)[2R_{1}-(2n-1)(g+s)]/C_{0}/2$$
(3)$$R_{SRRs}=\rho [R_{1}-(n-1)(g+s)]/(gt)$$
(4)$$L_{SRRs}=4\mu_{0}[R_{1}-(n-1)(g+s)][ln(0.98/\rho )+1.84\rho ]$$
where *n* = 3 is a ring number.

In the open resonant ring structure, the resonance effect is mainly generated by the outer ring. The inner ring is used to strengthen the coupling between the inner and outer rings so that the total capacitance of the open resonant ring structure is increased to achieve a shift of the resonant frequency to a lower frequency.

The inner split-ring resonator radius is *R_1_*, the middle split-ring resonator radius is *R_2_*, and the outer split-ring resonator radius is *R_3_* The inner, middle, and outer radii of the proposed radiation patch are proposed from [35].
(5)$$R_{3}=\frac{29\times 10^{6}}{f_{0}\sqrt{\varepsilon_{reff}}}$$
(6)$$R_{2}=0.69R_{3}$$
(7)$$R_{1}=0.38R_{3}$$
where *f_0_* is the lowest resonant frequency of the proposed antenna, and $\varepsilon_{reff}=\sqrt{\left(\varepsilon_{r}+1\right)/2}$ is the effective dielectric constant of the substrate.

To explain the motives for developing the multi-frequency antenna, we show the process of antenna design more precisely. It should be emphasized that it is a common method to realize multi-frequency antennas by using multi-loop structures. Therefore, we compare the results of several multi-loop antennas here. Figure 2 shows the structural diagram of the evolution of a single-band antenna into a multi-band antenna. During the design process, we also discussed the influence of the shift of the antenna back structure on the antenna performance, among which, the simulation results of the back without a ring, three rings and three cross-shaped open rings have a similar trend, but the cross-shaped open ring structure makes the overall performance of the antenna optimal. Therefore, in this paper, we focus on the impact of the upper surface radiation patch structure on antenna performance. Figure 3 shows the simulation reflection coefficient (S_11_) curve corresponding to the antenna evolution process.

As shown in Figure 3a, the radiant patch is an open resonator with a single ring, and the resonant frequency of the antenna is only 0.38 GHz. Figure 3b shows the double-rings radiant patch. Since multi-frequency resonances can be achieved between circular split ring resonators of different sizes, the resonant frequencies of the antennas are 0.25, 0.53, 1.06, 1.46, and 2.03 GHz, but only at 2.03 GHz, the return loss (S_11_) is less than −10 dB. Figure 3c based on the two becomes a three-ring open resonator radiation patch, enhancing the effect of multi-frequency resonance. Not only does the return loss (S_11_) at the resonant points at 1.10, 1.32, and 2.07 GHz exceed 10 dB but new resonant points are added at 1.63, 1.97, and 2.9 GHz, indicating that the antenna has a multi-frequency capacity and can operate simultaneously in six bands from 0 to 3 GHz.

To better understand the reason for the multi-frequency radiation properties of the antenna, the surface current distribution of the antenna was simulated and analyzed at the following six frequencies: 1.10, 1.33, 1.63, 1.97, 2.08, and 2.69 GHz. The results are shown in Figure 4.

As shown in Figure 4, the structure of the three open-loop resonators on the antenna surface. At lower frequencies, 1.10, 1.33, and 1.63 GHz. This suggests that, at lower frequencies, the outermost ring structure plays a major role in the emission. In the three higher frequency bands of 1.97, 2.08, and 2.69 GHz, the surface currents of the three open ring structures are uniform and high in density, so the three open ring structures act as radiation in the higher frequency bands.

Figure 5 shows the antenna radiation direction diagram in the E and H planes at six operating frequency points (1.10, 1.33, 1.63, 1.97, 2.08, and 2.69 GHz). As can be seen from Figure 5, when the antenna operates at 1.10, 1.33, and 1.63 GHz, surface E presents obvious “8” shaped half-wave dipole directional radiation characteristics, and the maximum radiation direction is near 0°. The orientation map of the surface H is approximately a positive circle, indicating that the antenna has omnidirectional radiation properties at this time. As the operating frequency changes from low to high, the direction of antenna radiation changes to some extent. At 1.97 GHz, the antenna radiates approximately in the direction of the half-wave dipole radiation, and the radiation direction shift is negligible. However, in the two high-frequency operating bands of 2.08 and 2.69 GHz, the current distribution of the split-ring resonator ring structure on the antenna surface is more complex. The surface currents between the different ring structures affect each other, resulting in a shift of the maximum radiation direction of the antenna to the left. The higher the frequency, the larger the deviation angle, especially at 2.69 GHz where the back-lobe appears in the radiation direction map and directivity is slightly affected. However, in all three frequency bands, the maximum deviation of the radiation direction is small, and the H-plane orientation map is still approximately a positive circle, so the antenna still has omnidirectional radiation properties. Figure 6 shows the gain of the antenna in the operating frequency band, which reaches a maximum gain of 4.63 dBi. Overall, the proposed antenna has excellent omnidirectional radiation properties at all operating frequency points, which will meet the needs of portable multi-band mobile devices.

## 3. Test and Analysis

To verify the performance of the antenna, we use an electromagnetic computer simulation (CST) microwave studio to simulate and analyze the antenna parameters. The actual return loss (S_11_) of the antenna is measured by an Agilent vector network analyzer under normal experimental conditions and compared with the simulation results. First, the microstrip line of the antenna is welded to the SMA joint matching the impedance of 50 Ω. Then, the network analyzer is connected to a coaxial cable with the same impedance matching and calibrated in the order of open circuit-short circuit-load. After the device calibration, the antenna is connected with a coaxial cable and tested. The measurement of the antenna is shown in Figure 7.

The comparison between antenna simulation and measurement results is shown in Figure 8. The experimental results show that six frequency points can reach below 10 dB, and the number of frequency points is consistent with the six frequency points of the theoretical simulation. The six frequency points measured in the experiment are 0.58 GHz, 0.93 GHz, 1.23 GHz, 1.49 GHz, 2.42 GHz, 2.61 GHz, and the theoretical simulation is 1.10 GHz, 1.33 GHz, 1.63 GHz, 1.97 GHz, 2.08 GHz, 2.69 GHz respectively. There are three reasons why these frequencies are not the same. The first reason is the size error brought by processing. The second reason is the limitation of experimental conditions, the measurements were not completed in the microwave darkroom, so the electromagnetic environment affected the antenna radiation. The third reason is that the SMA joint is not a perfect match with the sample.

## 4. Comparison and Discussion

As can be seen from Table 2, the proposed antenna structure can achieve more operating frequency points while having higher gain compared to existing multi-frequency antennas, but the operating bandwidth at each resonant frequency point is slightly narrower, so there is still room for improvement.

## 5. Conclusions

In this paper, we design and simulate a new multi-frequency omnidirectional microstrip antenna using CST simulation software. The split-ring resonator structure on the antenna surface and the defect ground state structure on the back of the antenna can effectively control the surface current distribution and increase the resonant frequency of the antenna, effectively increasing the number of operating frequency bands. The measurements are in general agreement with the simulations. The proposed antenna can work nicely in 5G NR (FR1, 0.45–3 GHz), 4GLTE (1.6265–1.6605 GHz), Personal Communication System (1.85–1.99 GHz), Universal Mobile Telecommunications System (1.92–2.176 GHz), WiMAX (2.5–2.69 GHz), and other communication frequency bands. The antenna also has excellent omnidirectional radiation properties in different operating frequency bands, which meet the requirements of portable multi-band mobile devices and are of practical engineering interest and utility.

## Figures and Tables

**Figure 1 micromachines-14-00994-f001:**
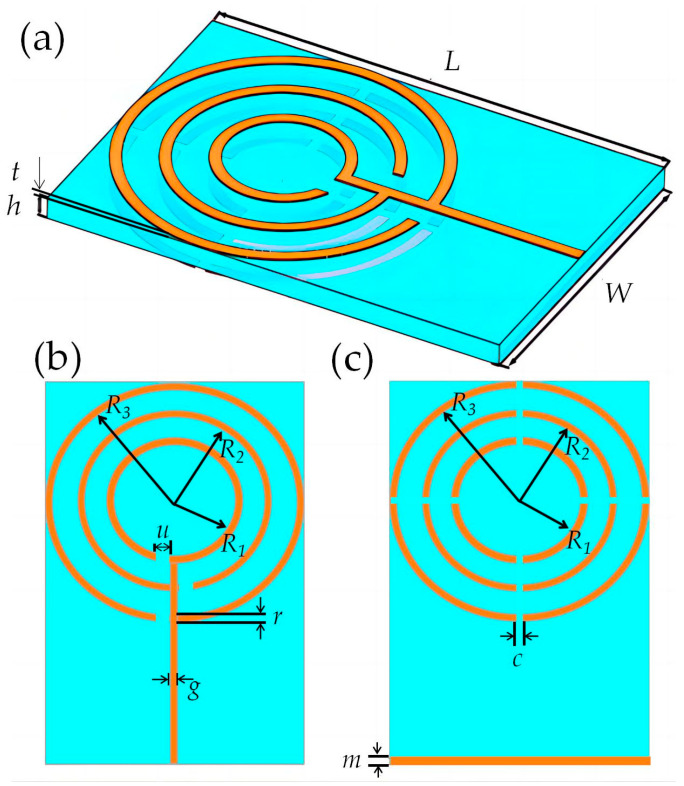
Antenna structure diagram. (**a**) Three-dimensional image of the antenna; (**b**) The front structure of the antenna; (**c**) The back structure of the antenna.

**Figure 2 micromachines-14-00994-f002:**

Antenna design process: (**a**) Single ring patch; (**b**) Double rings patch; (**c**) Three rings patch.

**Figure 3 micromachines-14-00994-f003:**
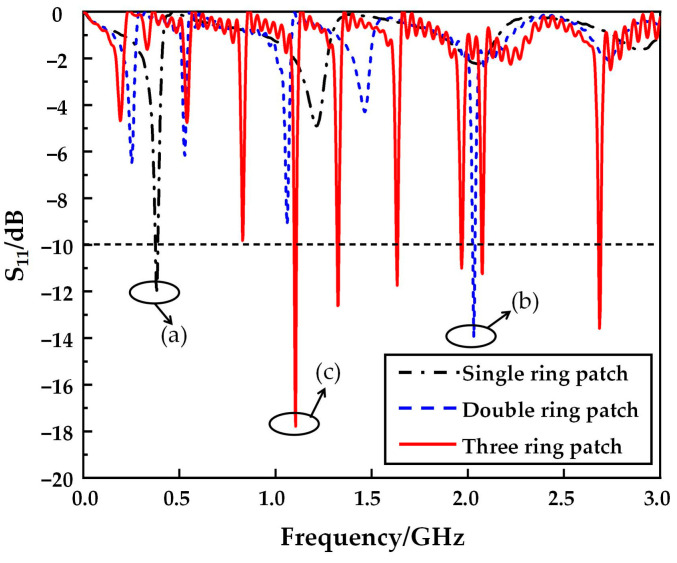
Simulation results of the return loss of three antennas. (a) Single ring patch; (b) Double rings patch; (c) Three rings patch.

**Figure 4 micromachines-14-00994-f004:**
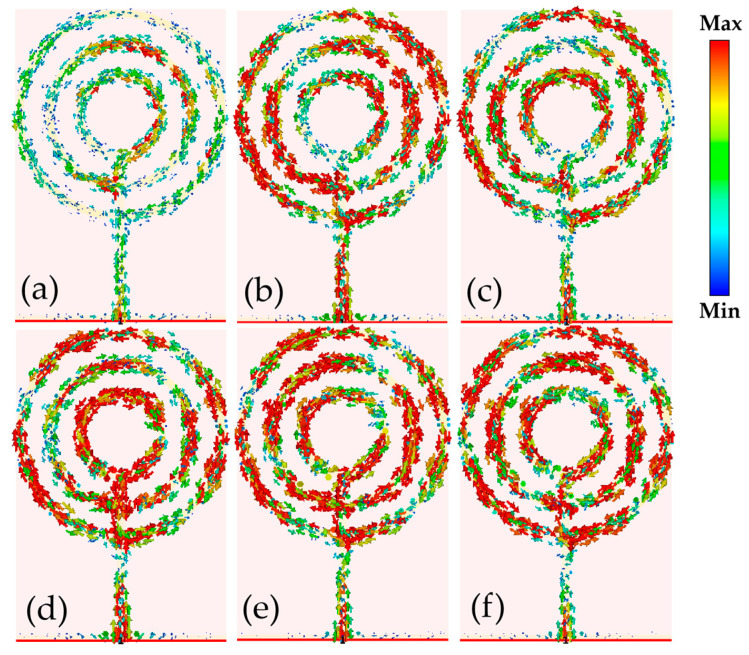
The surface current distribution of the antenna at (**a**) 1.10 GHz; (**b**) 1.33 GHz (**c**) 1.63 GHz; (**d**) 1.97 GHz; (**e**) 2.08 GHz; (**f**) 2.69 GHz.

**Figure 5 micromachines-14-00994-f005:**
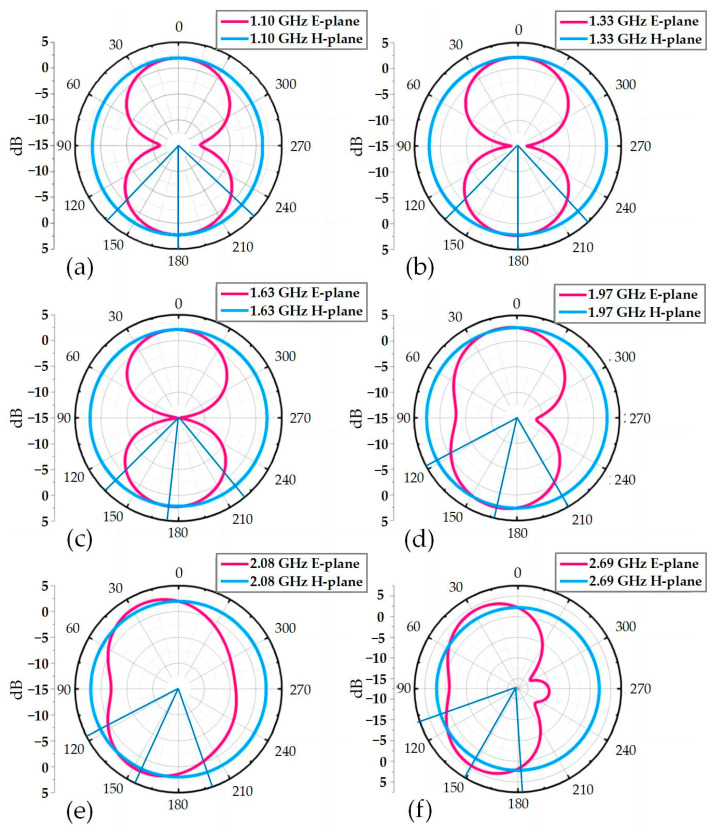
Antenna radiation direction diagram at (**a**) 1.10 GHz; (**b**) 1.33 GHz; (**c**) 1.63 GHz; (**d**) 1.97 GHz; (**e**) 2.08 GHz; (**f**) 2.69 GHz.

**Figure 6 micromachines-14-00994-f006:**
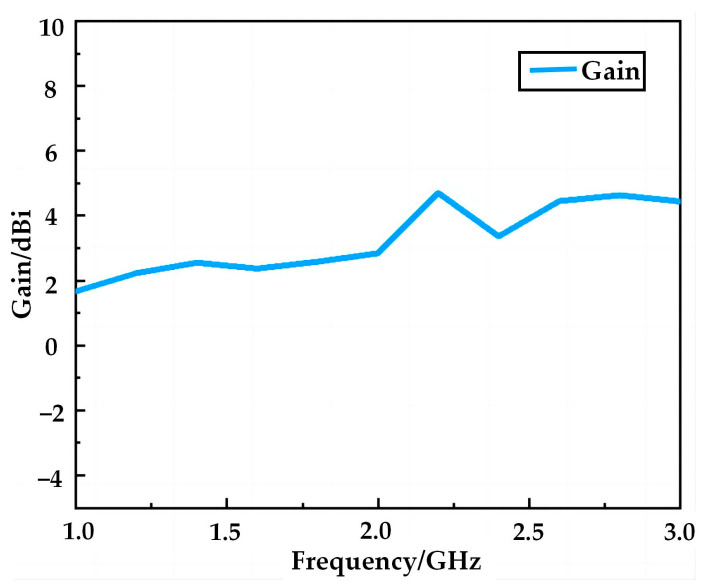
Simulation of antenna gain.

**Figure 7 micromachines-14-00994-f007:**
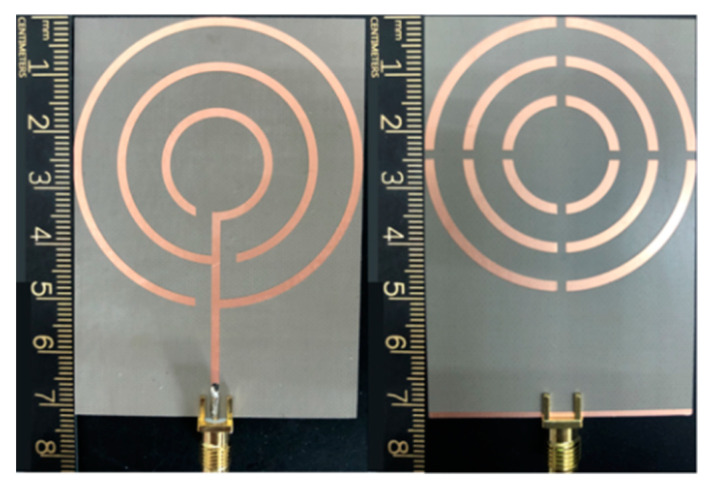
Photograph of the fabricated antenna.

**Figure 8 micromachines-14-00994-f008:**
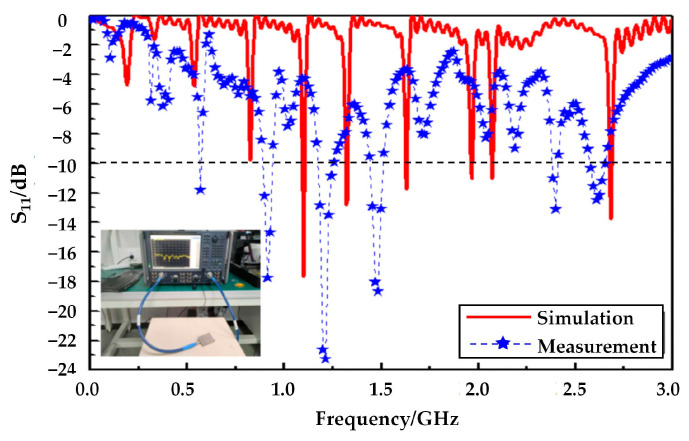
Simulated and measured reflection coefficient of the proposed antenna.

**Table 1 micromachines-14-00994-t001:** Design size of the multi-frequency antenna.

Parameter	*L*	*W*	*h*	*R_1_*	*R_2_*	*R_3_*	*u*	*r*	*g*	*c*	*m*	*t*
Value/mm	60	42	1.6	7.5	13.5	19.5	3.17	1.5	1.5	1.5	1	0.036

**Table 2 micromachines-14-00994-t002:** Research results of multi-frequency antennas.

Ref	Antenna Size Comparison (L × W mm^2^)	Resonance Frequency *f*_0_ (GHz)	The Number of −10 dB Bandwidth	−10 dB Bandwidth (GHz)	Maximum Gain
[6]	20 × 20	1.35/2.57/4.57/5.21	4	1.3–1.39/2.52–2.63/4.49–4.67/5.06–5.18	-
[15]	26 × 22	1.836/3.72/4.32	3	1.495–2.180/3.35–4.08/4.15–4.49	2.43 dBi
[20]	43.6 × 43.6	0.829/2.20/1.575/2.450	4	0.698–0.960/1.710–2.690/1.575/ 2.400–2.500	5.3 dBi
[22]	28.05 × 24.90	1.226/1.582	2	1.217–1.235/1.567–1.597	-
[24]	50 × 50	2.4/3.5/4.8	3	1.88–2.73/3.26–3.79/4.7–5.9	4.7 dBi
[25]	100 × 60	2.45/3.50/5.31	3	2.17–2.72/3.34–3.66/4.85–5.77	3.89 dBi
[27]	17 × 17	1.65/5.90	2	1.1–2.2/4.8–7.0	3 dBi
Proposed	62 × 42	1.10/1.33/1.63/1.97/2.08/2.69	6	1.097–1.109/1.319–1.329/1.628–1.635/1.964–1.971/2.071–2.078/2.679–2.691	4.63 dBi

## Data Availability

The data that support the findings of this study are available from the corresponding author upon reasonable request.

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
