# Peer review of "A Multi-Frequency Omnidirectional Antenna Based on a Ring-Shaped Structure"

_micromachines, 2023, doi:10.3390/mi14050994_

Round 1

Reviewer 1 Report

The specific paper attemtps to present a multi-frequency antenna based on ring-shaped structures. The topic seems interesting, however, the authors must explain the motives for developing their concept. Why do they have to select the specific structure?

The topic of the paper is definitely relevant in the field, yet its contribution is of medium interest. However, if the authors address the comments the paper could be accepted for publication.

1. The authors should explain the motives of developing such an antenna and the theory for the design.

2.  How the authors have extracted the simulation results? Please provide numerical evidence and the computational package.

3. The most important improvement would be the addition of the theoretical formulation. The authors start actually, with a very small section and go straight to the results. Moreover, they should be more clear with the use of the computational package and their simulations.

4. Some more real-world examples could be added. The inclusion of the comparison table is indeed a positive feature of the paper.

5. The references are appropriate.

6. The figures could be more clear, yet this could be a problem of my viewer. In any case, their overall appearance is adequate.

The English and the language of the manuscript are acceptable. There are several places in the Introduction that could be better expressed, yet the overall meaning is comprehensible.

Author Response

​Thank you so much for your review and comments. We have revised the article in detail and responded critically to your suggestions. Please refer to the attachments for details. Thank you again!

Reviewer 2 Report

The article lacks basic information about the adopted model for analysis, and what computer program was used. The table asks for the loss angle of the dielectric used.

You can see a careless approach to presenting the drawings, e.g. figure 3 is S11 and not S11

At the same time, there are large discrepancies between calculations and measurements, and the authors ignore this in silence

Figure 3 shows that the antenna will work properly on four sub-bands, although they show that it can work on six, see Figure 5.

Also, no information was given under what conditions the measurements were carried out.

No

Author Response

Thank you so much for your review and comments. We have revised the article in detail and responded critically to your suggestions. Please refer to the attachments for details. Thank you again!

Round 2

Reviewer 1 Report

The authors have addressed the majority of my comments.

Apart from some minor issues in the Introduction, the language and the structure of the revised paper seems ok.

Author Response

​Thank you so much for your comments and responses. We have responded to your comments in detail, see attached for details.

Reviewer 2 Report

There is still no information about the characteristics presented in Figure 5

Assuming that we are dealing with a normalized characteristic, the question is whether it is a power or voltage characteristic. And if so, what does crossing the "zero" level in Figures 5e and 5f mean

The conclusions are questionable, especially the statement that the antenna can work in the 5G system in the frequency range of 0.45 - 6 GHz

Author Response

(The authors gave the same response as above.)
